# Saponin Fractions from *Eryngium planum* L. Induce Apoptosis in Ovarian SKOV-3 Cancer Cells

**DOI:** 10.3390/plants12132485

**Published:** 2023-06-29

**Authors:** Małgorzata Kikowska, Hanna Piotrowska-Kempisty, Małgorzata Kucińska, Marek Murias, Jaromir Budzianowski, Anna Budzianowska, Mariusz Kaczmarek, Mariusz Kowalczyk, Anna Stochmal, Barbara Thiem

**Affiliations:** 1Laboratory of Pharmaceutical Biology and Biotechnology, Department and Division of Practical Cosmetology and Skin Diseases Prophylaxis, Poznan University of Medical Sciences, Collegium Pharmaceuticum, 3 Rokietnicka St., 60-806 Poznan, Poland; jbudzian@ump.edu.pl (J.B.); abudzian@ump.edu.pl (A.B.); bthiem@ump.edu.pl (B.T.); 2Department of Toxicology, Poznan University of Medical Sciences, Dojazd 30, 60-631 Poznań, Poland; hpiotrow@ump.edu.pl (H.P.-K.); kucinska@ump.edu.pl (M.K.); marek.murias@ump.edu.pl (M.M.); 3Department of Clinical Immunology, Poznan University of Medical Sciences, 5 Rokietnicka, 60-806 Poznań, Poland; markacz@ump.edu.pl; 4Department of Biochemistry and Crop Quality, Institute of Soil Science and Plant Cultivation, State Research Institute, 10 Czartoryskich St., 24-100 Puławy, Poland; mkowalczyk@iung.pulawy.pl (M.K.); asf@iung.pulawy.pl (A.S.)

**Keywords:** flat sea holly, saponin fractions, human ovarian cancer, apoptosis, caspase 3 activation

## Abstract

(1) The cytotoxicity and antioxidant activity of different fractions as well as the pro-apoptotic activity of saponin fractions from *Eryngium planum* L. in SKOV-3 was investigated. (2) In screening studies, the cytotoxicity of six fractions on SKOV-3 was examined by LDH and SRB assays. The most active fractions—triterpenoid saponins—were selected for further investigation. To determine the mechanism of saponin fractions’ cytotoxicity, their ability to induce apoptosis was examined via Annexin V assay. The effect of the saponin fractions on caspase 3 activity was measured using a Caspase 3 Assay Kit. The expression of 84 apoptosis-related genes was investigated in cancer cells exposed to saponin fractions from the roots. The radical scavenging capacity of different fractions was determined via DPPH assay. (3) The pronounced cytotoxic effects in SKOV-3 were demonstrated by saponin fractions from the leaves and roots. Those saponin fractions were chosen for further investigation. The treatment of cancer cell lines with saponins obtained from the roots provoked a significant increase in apoptotic cells. In the SKOV-3 cells, saponins caused upregulation of pro-apoptotic genes and a decrease in anti-apoptotic genes. The activation of caspase 3 was correlated with an increased DFFA expression level in the treated SKOV-3 cells. The most active fractions were phenolic acids from the shoots and roots. (4) To the best of our knowledge, the current study is the first to demonstrate that the barrigenol-type triterpenoid saponin fraction from the roots of *E. planum* inhibits SKOV-3 cell proliferation and induces apoptosis, which may be regulated by the expression of genes mostly specific to a mitochondria-related pathway.

## 1. Introduction

Ovarian cancer is the leading cause of death from gynecologic cancers; approximately 300,000 women are diagnosed with ovarian cancer each year, and there are over 180,000 deaths globally every year. The effect of current therapies is not satisfactory [1]. Therefore, many plant species have been examined to identify effective anticancer compounds and elucidate mechanisms of cancer prevention via apoptosis. Several studies have indicated that the saponins in plants are associated with anti-tumor effects on many cancer cells. Furthermore, saponins in combination with conventional tumor treatment strategies result in improved therapeutic success (reviewed in Man et al. [2]; Koczurkiewicz et al. [3]; Elekofehinti et al. [4]; and Podolak et al. [5]). 

*Eryngium planum* L. (the Saniculoideae subfamily of the Apiaceae family) is of great value for use in traditional European medicine because it contains triterpenoid saponins, phenolic acids, flavonoids, coumarin derivatives, essential oils, and acetylenes [6]. 

Triterpenoid saponins are a large class of natural compounds that exhibit a wide variety of structural diversity and biological activity. They have a broad range of pharmacological applications, such as hypocholesterolemic, immunoadjuvant, antiviral, antibacterial, antifungal, anti-leishmanial, and anti-inflammatory applications. Saponins also exhibit anti-HIV-1 protease activity, genotoxicity, cytotoxicity, and toxicity-enhancing properties against cancer cell lines [2,3,4,5,7]. The majority of *Eryngium* saponins belong to polyhydroxylated oleanane triterpenoid saponins [6].

*E. planum* organs of wild-grown plants contain a complex of triterpenoid saponins. Three main triterpenoid saponins were isolated and their tentative identifications (the MS/MS fragmentation pattern) were confirmed by 1D and 2D NMR analyses [8] (Table 1). 

According to our previous studies, the main three ES1, ES2, and ES3 triterpenoid saponins of *E. planum* are derivatives of acylated R1 and A1-barrigenols: 3-*O*-*β*-D-glucopyranosyl-(1→2)-*β*-D-glucuronopyranosyl-21-*O*-acetyl,22-*O*-angeloyl-R1-barrigenol, 3-*O*-*β*-D-glucopyranosyl-(1→2)-*β*-D-glucuronopyranosyl-22-*O*-angeloyl-R1-barrigenol, and 3-*O*-*β*-D-glucopyranosyl-(1→2)-*β*-D-glucuronopyranosyl-22-*O*-angeloyl-A1-barrigenol [8]. 

Saponins (ES4-ES6) occurring in very low concentrations are not yet fully characterized, although based on MS/MS fragmentation, their aglycones are different from those already identified and described in the literature [6,8]. Analyses of the structure–activity relationship have revealed that acylation with angeloyl groups at the C21 or C22 position and carbohydrates attached to the C3 position in triterpenoid saponins may be crucial for the cytotoxic and pro-apoptotic effects of those compounds [9]. 

The aim of the present study was to select the most cytotoxic fraction from *Eryngium planum* L. and then examine the induction of apoptosis and measure pro- and anti-apoptotic gene expression in ovarian SKOV-3 cancer cells. 

## 2. Results

The preliminary phytochemical analyses of extracts from *E. planum* revealed that rosette leaves produced flavonoid compounds, phenolic acids, and several saponins and the roots accumulated phenolic acids and saponins. Qualitative analyses confirmed the presence of selected phenolic acids: rosmarinic, chlorogenic, and caffeic and triterpenoid saponins—derivatives of acylated R1 and A1-barrigenols [8,10].

In order to evaluate the cytotoxic and pro-apoptotic effects of those groups of compounds, the extracts were fractioned, and fractions from the rosette leaves (flavonoid (1), flavonoid-saponin (2), saponin (3), and phenolic acid (4) fractions) and from the roots (saponin (5) and phenolic acid (6) fractions) were examined.

### 2.1. Effect of the Fractions on SKOV-3 Viability

The inhibitory effect of the six fractions from the rosette leaves and roots of *E. planum* tested against the SKOV-3 human cancer cell line was evaluated by LDH and SRB assays. Among all of the tested fractions at different concentrations (100–1 µg mL^−1^), the pronounced cytotoxic effects after 24 h treatment of cancer cells were demonstrated by saponin fractions from the leaves and roots at concentrations of 25 and 50 µg mL^−1^ (Figure 1). Those saponin fractions were chosen for further investigation. Qualitative LC-MS analyses revealed that the rosette leaves and roots contain entirely different sets of saponins (Figure 2). Quantitative analyses of saponins isolated from the roots indicate that these compounds are indeed root-specific and their occurrence in rosette leaves is incidental (Figure 2). There were two saponins in the rosette leaves in an amount of 0.185 ± 0.01 mg g^−1^ DW and six saponins in the roots in an amount of 5.626 ± 0.02 mg g^−1^ DW (Figure 2, Table 2). 

### 2.2. Effect of Saponin Fractions on Apoptosis in SKOV-3

The induction of apoptosis was assayed via an Annexin V-FITC Apoptosis Detection Kit. The statistically significant pro-apoptotic effect was noticed for the saponin fraction from the roots at a conc. of 50 µg mL^−1^. The number of necrotic cells was also determined—statistically significant differences between the control and saponin fraction from the roots at a conc. of 25 µg mL^−1^ were not found, but at a conc. of 50 µg mL^−1^, they were observed (Figure 3A). The saponin fraction from the leaves showed a moderate pro-apoptotic effect (Figure 3B).

### 2.3. Effect of the Saponin Fractions on Caspase 3 Activity in SKOV-3

The effect of the saponin fractions (25 and 50 µg mL^−1^) from the leaves and roots on the caspase 3 activity in the SKOV-3 cancer cell line was measured using a Caspase 3 Assay Kit. The activation of caspase 3 occurred in cells treated with both fractions at both concentrations (Figure 4). The statistical differences were demonstrated for fractions from different organs and by the lack of these differences for fractions at different concentrations. Statistically significant pro-caspase 3 activation was observed for the saponin fractions from the roots. Therefore, this fraction at a conc. of 50 µg mL^−1^ was chosen for further investigation. 

### 2.4. Effect of the Saponin Fraction from the Roots on the Expression of Apoptosis-Related Genes

To elucidate the mechanism by which the saponin fraction from the roots induced apoptosis, the expression of pro- and anti-apoptotic genes in the SKOV-3 line was investigated. Significant changes in the expression profile of apoptotic genes were found. After treatment of the cancer cells with 50 µg mL^−1^ with the saponins, the expression of 10 pro-apoptotic genes (*BAX*, *BID*, *HRK*, *TP53*, *PMAIP1*, *DFFA*, *CASP3*, *CASP 9*, *TNFRSF10A*, and *TNFRSF10B*) increased (Figure 5) and that of 4 anti-apoptotic genes (*BCL2L10*, *TNFRSF10C*, *TNFR2*, and *TRAF7*) decreased (Figure 6). 

### 2.5. Analysis of the DPPH Scavenging Activity of the Extract Fractions

The most widely used spectrometric technique for evaluating the ability of antioxidants (e.g., extracts, fractions, and isolated compounds) to scavenge free radicals—the DPPH assay—revealed that the most active fractions were phenolic acids from the shoots and roots, with percentages of DPPH reduction of 62.37 ± 0.13 and 80.84 ± 0.08, respectively, and IC_50_ values of 76.81 ± 0.18 µM and 34.55 ± 0.05 µM, respectively (Table 3). 

## 3. Discussion

In this study, a screening analysis of six fractions from *Eryngium planum* L. (from the rosette leaves: flavonoids, flavonoid-saponin, saponin, and phenolic acid fractions, and from the roots: saponins and phenolic acid fractions) on the SKOV-3 line was performed. As a result, the most cytotoxic fractions were selected—saponins from the leaves and roots.

Numerous papers reporting the biological activity of saponins [11] have pointed to their anticancer properties, which have been reviewed separately [5,12], including barrigenol-type derivatives (about 360 compounds classified into five subtypes: barrigenols A1, A2, R1, C, and 16-deoxybarringtogenol C) [13,14], which are represented by saponins (barrigenols A1 and R1) of the genus *Eryngium* [6,8].

Among the many biological activities of the barrigenol saponins potentially valuable due to their pharmacological applications, their anti-tumor activity has attracted special attention [14]. In general, this type of activity is due to cell cycle arrest, the induction of apoptosis, the inhibition of DNA-topoisomerase I, and the inhibition of cell migration [14].

Podolak et al. [5] broadly analyzed the structural features influencing the anti-tumor activity of oleanan-type saponins. It was summarized, that the following features are important for the cytotoxic activity: a double bond, a carboxylic group (COOH) at C-28 preferably without substitution with sugars, the number and position of hydroxyl (OH) groups, sugar substitution (particularly with α-L-rhamnose), and acylation (e.g., with an angeloyl group). Nevertheless, it was difficult to find a straightforward rule for the structure–activity relationship (SAR).

Wang et al. [15] determined the structure–anticancer activity relationship for barrigenol saponins through an analysis of 25 compounds by using the 3D-QSAR (three-dimensional quantitative structure–activity relationship), CoMFA (comparative molecular field analysis), and CoMSIA (comparative molecular similarity index analysis) methods. The key structural features found were: ester-bonded angeloyl groups at C-21 and C-22, OH groups at C-15, C-16, C-21, C-22, and C-28, a sugar chain at C-3, and the absence of a long sugar chain at C-28. Of the mentioned structural groups, C-22-angeloyl, OH groups at C-15, C-16, C-21, and C-28, and a sugar group at C-3 occur in the saponins isolated from *E. planum* [8].

It is believed that at least one angeloyl moiety at either the C-21 or C-22 position makes a significant contribution to the cytotoxicity, but the type and number of sugar moieties at C3 may decrease their cytotoxic effect. Chan proved that acetylation with angeloyl groups at the C21 and C22 positions in triterpenoid saponins is essential for cytotoxicity toward tumor cells. The cell growth inhibition activity in OVCAR3 cells was determined by MTT assays—compounds with one angeloyl group (at either C21 or C22) have reduced activity as compared to those with two angeloyl groups; the removal of both groups completely abolishes the activity. Moreover, it was also noticed that when carbohydrates were removed from the C3 position of R1-barringenol-type saponin, it also lost activity [9].

Due to the interesting profile of secondary metabolites (mainly triterpenoid saponins) of species belonging to the *Eryngium* genus, in the world literature, a number of studies have reported the cytotoxic activity of those taxa. The cytotoxicity effect of extracts and crude saponins has been reported for *E. planum*, *E. creticum*, *E. maritimum*, *E. kotschyi*, *E. yuccifolium*, and *E. campestre* [16,17,18,19,20]. It was noted that the 1301 and HL60 cell lines had a viability of 62 and 55% (trypan blue assay), respectively, after 24 h treatment with ethanolic extract (300 µg mL^−1^) from *E. planum* fruits [16]. The results of the XTT technique showed that the methanolic extract of the *E. creticum* leaves and stems inhibited the growth of MCF7 by 72% and 68%, respectively [17]. The cytotoxic activity of the lyophilized aqueous aerial and root parts of *E. maritimum* and *E. kotschyi* has been investigated on various cell lines. Inhibitory concentration values in most cases vary around 16.33–125.66 µg mL^−1^. The highest cytotoxicity on Heg2 cells measured by the MTT assay was found for the roots of *E. maritimum* (30.25 µg mL^−1^) and *E. kotschyi* (32.86 µg mL^−1^) [18]. The saponin from *E. yuccifolium* exhibited moderate cytotoxicity against A549, PC-3, HL-60, and MRC-5 cell lines, while no toxicity against the human pancreas cancer cell line PANC-1 was shown [19]. Five saponins from *E. campestre* showed weak cytotoxic activity against HCT 116 and HT-29 human tumor cells [20]. In view of the research findings and our observations, we decided to include saponins from the leaves and roots for further investigation.

In the present study, the cytotoxicity effect of saponins from *E. planum* correlated with an increased level of apoptosis. The significant pro-apoptotic effect on SKOV-3 was noticed for the roots’ saponin fraction (50 µg mL^−1^). Similarly, the ethanolic extract (300 µg mL^−1^) from *E. planum* fruits was found to induce apoptosis in two human leukemic cell lines C8166 and J45 after 24 h incubation. A moderate effect was observed in HL-60 and ML-1 cell lines [16]. The positive effects of triterpenoid saponins on proapoptotic activity have been well documented in the literature. The saponin fraction (40 µmol mL^−1^) and five individual saponins from *Anemone flaccida* significantly affected the percentage of apoptosis in HeLa cells [21]. Moreover, the saponin-rich extract from the roots of *Gypsophila oldhamiana* influenced the induction of programmed cell death in SMM-7721 [22]. The saponin fraction of the roots from *Platycodon grandiflorum* induced apoptosis in HT29 cells [23].

Several authors have suggested that cancer-therapy-induced apoptosis involves the activation of the Fas receptor/ligand system; factors other than drugs induce apoptosis by initiating the release of cytochrome c from mitochondria. Therefore, the analysis of the expression profile of genes driving mitochondria- and receptor-mediated apoptosis pathways may contribute to the elucidation of the mechanism of saponin fraction anticancer action. In the present study, it was shown that the saponin fraction from the roots of *E. planum* caused upregulation of *BAX*, *BID,* and *HRK* mRNA in the SKOV-3 cells, which are members of the *BCL-2* gene family, and TP53, a protein-p53-coding gene. Moreover, the expression of *PMAIP1* demonstrated an increased level in the treated SKOV-3 cells. *BAX* and *PMAIP* have been shown to be involved in p53-dependent apoptosis [24]. It is likely that p53 promotes *BAX*’s apoptotic faculties in vivo as a primary transcription factor. However, p53 also has a transcription-independent role in apoptosis [25]. In the analyzed experiment, the saponin fraction from the roots also caused up-regulation of *DFFA*, *CASP3,* and *CASP9*. The apoptotic process is accompanied by shrinkage and fragmentation of the cells and nuclei and degradation of the chromosomal DNA into nucleosomal units. *DFFA* encodes the DNA fragmentation factor, which is a substrate for caspase 3 and triggers DNA fragmentation and chromatin condensation during apoptosis [26]. In the intrinsic activation of caspase 3, cytochrome c from the mitochondria works in combination with caspase 9 to process procaspase 3 [27]. This correlated with our finding that the activation of caspase 3 occurred in SKOV-3 treated with saponin fractions from *E. planum*. Our results, which show an increased level of *BAX* and *CASP9* in SKOV-3 after saponin treatment correlate with the reports of research conducted by Kim et al. [28]. An analysis of apoptotic gene expression in human colorectal carcinoma cells (HCT-15) after treatment with a crude mixture of saponins from *Panax ginseng* C.A. Meyer revealed a high level of proapoptotic genes, including those encoding caspase 9 and Bax. The authors suggested that saponins from ginseng induce apoptosis of the mitochondrial pathway [28]. This is in agreement with the studies of Fang et al. [29], who investigated the effect of total saponins from *P. ginseng* on apoptosis in HL 60 cells. The results showed that the expression of apoptosis-related genes in cancer cells changed after treatment with saponins—*BAX* mRNA increased, and *BCL-XL* mRNA decreased. The authors concluded that the effect on the upregulation of BAX and the downregulation of *BCL-XL* probably play an important role in the apoptosis of HL 60 cells induced by saponins from ginseng [29], while studies by Jeong et al. [30] demonstrated that treatment of human monocyte-like histiocytic cells (U937) with Akebia saponin D from *Dipsacus asper* Wall induced apoptosis. Significantly increased expression of *TP53* and *BAX* genes was recorded [30]. Hsu et al. showed that after the addition of saikosaponin d, one of the active terpenes from *Bupleurum falcatum* L., to human CEM cells, the level of *TP53* mRNA significantly increased and the level of *BCL-2* reduced. The authors suggested that apoptosis induced by saikosaponin d in CEM lymphocytes may be partly mediated by *TP53* and *BCL-2* gene deregulation [31]. In this study, *BCL2L10* expression in SKOV-3 cells decreased after treatment with saponins. The protein encoded by this gene has been shown to suppress cell apoptosis possibly through the prevention of cytochrome C release from the mitochondria [32]. Surprisingly, we found a significant increase in *TNFRSF10A* and a moderate increase in *TNFRF10B* transcripts’ expression. Activated *TNFRSF10A* and *TNFRSF10B* transduce cell death signals and induce cell apoptosis. The receptor encoded by *TNFRSF10C* is not capable of inducing apoptosis and is thought to function as an antagonistic receptor that protects cells from TRAIL-induced apoptosis [33].

Most scientific papers on the biological activity of extracts from the organs of *Eryngium* species concern their antioxidant activity. Over the years, a number of research methods have been developed that allow for the determination of the antioxidant potential of extracts, fractions, or compounds isolated from plants. Both in vivo and in vitro tests can be distinguished here. In vitro methods are easier to perform, faster, and less expensive; they are usually characterized by high reproducibility of results. One of the most frequently used methods is the method using the DPPH (2,2-diphenyl-1-picrylhydrazyl) radical solution, thanks to which we can test the ability to scavenge free radicals [34]. The antiradical activity of aqueous and ethanolic extracts from the whole plant of *E. planum* was measured by the DPPH test, and the IC_50_ values were 1.731 ± 0.12 mg mL^−1^ and 1.362 ± 0.11 mg mL^−1^, respectively [35]. In another study, methanol and water-methanol (50%) extracts from *E. planum* fruits were tested for their overall antioxidant potential in the β-carotene/linoleic acid system. The study showed that the water–methanol extract (EC_50_ = 4.51 mg mL^−1^) showed stronger activity than the methanol extract (EC_50_ = 8.64 mg mL^−1^). Trolox, which is a synthetic water-soluble vitamin E derivative with high antioxidant activity, used as a reference compound, showed an EC_50_ value of 0.89 mg mL^−1^ [36]. An extract from the aerial part of *E. campestre* (a related species native to Poland) as well as isolated flavonoid compounds of the flavonol class (at a concentration of 0.5 mg mL^−1^) were tested for their antioxidant activity. The activity (DPPH test) was relatively high for the crude extract (66.3%) and exceeded the activity of all tested flavonoids [37]. In another study, the antioxidant activity of *E. campestre* extracts was analyzed by various techniques. Using the DPPH method, the ethanol extract from the roots of the holly root showed higher activity (IC_50_ = 0.70 mg L^−1^) than the extract from the above-ground part of the plant (IC_50_ = 1.14 mg L^−1^). The activity of both extracts slightly differed in the TBARS test (above ground 50%; roots 44%). In addition, both ethanol extracts showed low activity in the test of oxidative degradation of β-carotene and inhibition of lipid peroxidation (TBA test) [38]. In another study, the antioxidant properties of various extracts (extractants—butanol, water, and ethyl acetate) from both the roots and aerial parts of *E. campestre* were tested using the DPPH method and the β-carotene/linoleic acid system. According to the study results, the aerial butanol extract showed the lowest IC_50_ value (16.140 μg mL^−1^) [39]. However, the butanol extract from the root of *E. maritimum* (a related species native to Poland) was the most active in the DPPH test with an IC_50_ value of 0.0818 ± 0.0048 mg mL^−1^ and in the method of determining the iron ion reducing capacity (FRAP) with the IC_50_ = 0.1113 ± 0.0112 mg mL^−1^ [40]. In turn, other authors used the xanthine oxidase and DPPH test to assess the antioxidant activity of extracts from the above-ground part of *E. maritimum*. The results of two different antioxidant techniques showed that the aqueous extracts showed the best response in the DPPH test (>70%), while the ethanol extracts showed the best results in the xanthine oxidase test [41]. These tests showed the different activity of the tested samples; however, due to the lack of standardization and the variants of the methods differing in the conditions of the analysis, it often makes it impossible to compare the results.

## 4. Materials and Methods

### 4.1. Plant Material

Rosette leaves and roots of wild *E. planum* plants were collected from natural habitats in Poland (Łukaszewo, Kuyavian-Pomeranian province) in August 2018. The voucher specimens (no. EP-2008008-01K) were deposited in the Herbarium of the Medicinal Plant Garden at the Institute of Natural Fibres and Medicinal Plants in Poznań (Poznań, Poland).

### 4.2. Determination of Triterpenoid Saponins in Plant Material

Samples of plant material—rosette leaves and roots of *E. planum* (approx. 100 mg)—were mixed with diatomaceous earth, loaded into stainless steel extraction cells, and extracted with 80% (*v*/*v*) methanol using an accelerated solvent extraction system (ASE 200, Dionex, Sunnyvale, CA, USA). Extractions were carried out at a 10 MPa operating pressure at 40 °C. The evaporated to dryness under reduced pressure extracts were reconstituted in 2.0 mL of methanol containing 0.05% (*w*/*v*) ascorbic acid. The samples were then stored at −20 °C, and immediately before the analyses, they were diluted 20 times with doubly distilled water and centrifuged at 23,000× *g* for 15 min. Quantitative analyses were performed on a Waters Aquity UPLC-MS system (Waters, Milford, MA, USA) equipped with a triple quadrupole mass spectrometer (Waters TQD). The analytes were separated on a Waters BEH C18 column (100 × 1 mm, 1.7 µm) using a linear 6 min-long gradient from 5 to 80% of acetonitrile containing 0.1% (*v*/*v*) formic acid (solvent B) in 0.1% formic acid (solvent A) with the flow of 140 µL min^−1^. Separations were carried out at 50 °C. The column was washed with pure solvent B for 2 min and re-equilibrated with 5% solvent B in solvent A for 7 min prior to each injection. The injections were carried out in the ‘partial loop needle overfill’ mode of a Waters Aquity autosampler. One µL was injected from each sample, and the analysis of each sample was repeated three times. Column’s effluent was introduced into the ion source of the mass spectrometer, which operated in the negative ion mode with the following parameters of the ion source: cone voltage 130 V, capillary voltage 3.1 kV, extractor 3 V, RF lens 100 mV, source temperature 120 °C, desolvation temperature 350 °C, desolvation gas flow 500 L h^−1^, cone gas flow 50 L h^−1^, and collision gas flow 100 µL min^−1^. The collision cell entrance was set to −2 and the exit was set to 0.5. The parameters of quadrupoles 1 and 3 were set to achieve unit–mass resolution: both LM and HM resolutions were set to 15, and the ion energies were set to 0.9. The quantitation method was calibrated from the set of standard solution dilutions in the range of 100 pg to 16 ng µL^−1^. The triterpenoid saponins (eryngium saponins) were analyzed in the single ion monitoring mode, in which deprotonated quasi-molecular ions at *m/z* 925 (ES1), 967 (ES2), 909 (ES3) 1251 (ES4), 907 (ES5), and 895 (ES6) were observed. The obtained data were processed using Waters MassLynx version 4.1 SCN 714 software.

Three primary saponins were previously identified based on the electrospray MS/MS fragmentation and confirmed by 1D and 2D NMR analyses. NMR was performed on a Bruker DRX-600 spectrometer (Bruker BioSpin GmBH, Rheinstetten, Germany) equipped with a Bruker 5 mm TCI CryoProbe at 300 K. All 2D NMR spectra were acquired in CD3OD (99.95%, Sigma-Aldrich, St. Louis, MO, USA) and standard pulse sequences and phase cycling were used for DQF-COSY, HSQC, HMBC, and ROESY spectra. The NMR data were processed using UXNMR software [8].

### 4.3. Preparation of the Plant Extract Fractions for Bioactivity Assays

Dried and powdered rosette leaves of *E. planum* (82.8 g) were extracted with boiling 70% ethanol (4 × 600 mL). The extracts were combined and concentrated under reduced pressure to give a dry extract (32.4 g). A portion of the dry extract (27.7 g) was separated over a polyamide MN-6 (Macherey-Nagel; grain size 0.05–0.16 mm) column by elution with water, 20%, 40%, 60%, 80%, and 100% methanol, and methanol with 0.01% ammonia. Column fractions were combined on the basis of the results of the TLC examination to give the sugar fraction, coumarin fraction, flavonoid fraction, flavonoid-saponin fraction, saponin fraction, and phenolic acid fraction. Dried and powdered roots of *E. planum* (351.0 g) were extracted with boiling 70% ethanol (4 × 3 L). The combined extracts were evaporated to give a dry extract (145.5 g), a portion of which (136.0 g) was separated over a polyamide column eluted with water, 100% methanol and 0.01% ammonia in methanol. The column fractions were combined after TLC examination to give the sugar fraction, saponin fraction, and phenolic acid fraction.

For the detection of phenolic acids and flavonoids, 5 µL aliquots of each fraction were applied to the cellulose and HPTLC silica gel plates (10 × 20 cm, Merck, Darmstadt, Germany), and the plates were developed in ethyl an acetate–acetic acid–water (8:1:1 *v*/*v*/*v*) mixture. The plates were viewed under UV_366_ nm or daylight, before and after spraying with (i) NA (Roth) 0.1% solution in ethanol for phenolic acid detection (blue bands) and flavonoids (yellow bands) and (ii) AlCl_3_ 1% solution in ethanol for the detection of flavonoids. For the detection of saponins, 5 µL aliquots of each fraction were applied to the HPTLC silica gel plates (10 × 20 cm, Merck, Germany), and the plates were developed with a 1-buthanol–acetic acid–water 4:1:5 (*v*/*v*/*v*) mixture. The plates were viewed in daylight after spraying with vanillin-sulfuric acid reagent. The spots with a violet-pink color were considered as saponins. For the detection of coumarins, 5 µL aliquots of each fraction were applied to the HPTLC F 254 nm silica gel plates (10 × 20 cm, Merck, Germany), and the plates were developed with toluene–methyl ethyl ketone 9:1 (*v*/*v*). The spots were fluorescent blue.

The fractions from the rosette leaves the flavonoid (1), flavonoid-saponin (2), saponin (3), and phenolic acid (4) fractions, and those from the roots: the saponin (5) and phenolic acid (6) fractions were subjected to examination.

The stock solutions of the tested fractions were prepared in DMSO (Sigma-Aldrich Co., St. Louis, MO, USA) at a concentration of 1 mg mL^−1^.

### 4.4. Cell Culture

The SKOV-3 ovarian cancer cell line was purchased from the European Type Culture Collection (ECACC, Salisbury, UK), cultured in phenol-red-free DMEM (Sigma-Aldrich Co., St. Louis, MO, USA), and supplemented with 10% fetal bovine serum (FBS), 2 mM glutamine, penicillin (100 U/mL) and streptomycin (0.1 mg mL^−1^) (Gibco Invitrogen Corp., Grand Island, NY, USA) at 37 °C in a humidified atmosphere containing 5% CO_2_.

### 4.5. Cell Viability Assays

To investigate the effects of the plant fractions on cell viability, a lactate dehydrogenase (LDH) assay and a sulforhodamine B (SRB) cytotoxicity assay were performed. The cytotoxic effect of the tested fractions was evaluated by measuring cell membrane integrity using an LDH assay kit (Pointe Scientific; Canton, MI, USA). The SKOV3 cells were seeded in 96-well plates at a density of 2 × 10^4^ cells/well in 100 µL of growth medium and incubated under cell culture conditions. Then, the tested fractions from the rosette leaves: the flavonoid, flavonoid-saponin, saponin, and phenolic acids fractions and those from the roots: saponins and phenolic acid fractions were added at the following concentrations: 100 µg mL^−1^, 10 µg mL^−1^, and 1µg mL^−1^. DMSO was used as a control and Triton X-100 (Sigma-Aldrich Co., St. Louis, MO, USA) at a concentration of 1% in the culture medium was used as a positive control. The cells were incubated for 24 h, and 50 µL medium from each well was transferred to a new plate and 150 µL of LDH solution (NAD^+^, L-lactate, TRIS buffer) was added to each well. The plates were agitated on an orbital shaker for 10 min and the absorbance was read at 340 nm using a microplate reader Elx-800 (BioTek, Winooski, VT, USA). The cell viability was measured using an SRB assay. The SKOV-3 cells were seeded in 96-well plates at a density of 2 × 10^4^ cells per well, and the cells were allowed to attach overnight. Subsequently, the saponin fractions from the leaves and roots at concentrations of 100, 75, 50, 25, 12.5, 6.25, and 1µg mL^−1^ were added, and the cells were incubated for 24 h under cell culture conditions. DMSO was used as a control, and the concentration did not exceed 0.1% in the culture medium. After incubation, the cells were fixed with 200 µL of 10% trichloroacetic acid (Sigma-Aldrich Co., St. Louis, MO) for one hour at 4 °C. Then, the cells were washed five times with deionized water and incubated with 200 µL SRB (0.4% in 1% acetic acid) for 30 min at room temperature. The excess dye was removed by washing repeatedly with 1% acetic acid. The protein-bound dye was dissolved with 200 µL of 10 mM Tris base (Sigma-Aldrich Co., St. Louis, MO, USA) solution. The absorbance was measured at 540 nm using a microplate reader Elx-800 (BioTek, USA). The concentration of plant fractions that is required for 50% cell growth inhibition (IC_50_) was determined from a plot of percent cell viability versus the logarithm of concentration.

### 4.6. Assessment of Apoptosis

According to the manufacturer’s instructions, apoptosis and necrosis induction by saponin fractions was determined by an Annexin V-FITC Apoptosis Detection Kit (Caymanchem, An Arbor, MI, USA). Briefly, the cells were seeded in a 6-well plate at a density of 5 × 10^5^ cells per well for 24 h, and the saponin fractions were seeded at concentrations of 25 µg mL^−1^ and 50 µg mL^−1^. DMSO was used as a control, and the concentration of DMSO did not exceed 0.1%. The cells were treated with the tested fractions for 4 h. Subsequently, the cells were collected by trypsinization, and the apoptotic and necrotic cells were determined by annexin V/propidium iodide double staining and analyzed by flow cytometry (Beckton Dickinson FACScanTM; Beckton Dickinson; Franklin Lakes, NJ, USA).

### 4.7. Caspase 3 Activity

Caspase 3 activity was determined using a fluorimetric assay kit (Caspase 3 Assay Kit, Sigma-Aldrich Co., St. Louis, MO, USA) according to the manufacturer’s protocol. The SKOV-3 cells were seeded in a 96-well plate at a density of 2 × 10^4^ cells per well, and after 24 h, they were treated with saponin fractions at concentrations of 25 and 50 µg mL^−1^ and incubated for 4 h under cell culture conditions. DMSO was used as a control. The fluorescent product 7-amino-4-methylcoumarin (AMC) was measured using a microplate reader Tecan Infinity 200 (Männedorf, Switzerland) (excitation and emission wavelengths of 360 and 460 nm). The activity of caspase 3 was calculated using the AMC standard curve.

### 4.8. PCR-Array Analysis

After 4 h treatment of the saponin fraction from the roots (50 µg mL^−1^), the cells were harvested, and total RNA was isolated using a single-step method through acid guanidinium thiocyanate-phenol-chloroform extraction [42]. The RNA concentration was quantified by measuring the optical density at 260 nm. The RNA samples were treated with DNase I and reverse-transcribed into cDNA using oligo-qT primers. The PCR-array was carried out on a LightCycler^®^ Instrument 480 MultiwellPlate96 (Roche, Mannheim, Germany) using a LightCycler^®^ 480 Probes Master kit. The target cDNA was quantified using the relative quantification method. RealTime ready Human Apoptosis Panel 96 (Roche, Mannheim, Germany) was used to conduct the PCR-array analysis. For amplification, 20 µL cDNA solution was added to 980 µL of LightCycler^®^ 480 Probes Master (Roche) and then transferred in a 10 µL volume to each primer- and probe-based well of the 96-well format real-time PCR. The quantity of 84 pro- and anti-apoptotic genes was standardized by seven housekeeping genes.

### 4.9. DPPH Scavenging Activity

The free radical scavenging activity of the fractions was determined using the DPPH (1,1-diphenyl-2-picrylhydrazyl) standard method according to Molyneux [43]. The column chromatography fractions and a reference compound—quercetin (Lachema, Brno, Czech Republic)—were dissolved in methanol (200 µg mL^−1^ each). One milliliter of the fraction or the reference solution dilution (final concentrations of 1, 2.5, 5, 10, 20, 40, 80, and 100 µg mL^−1^) was mixed with 1.0 mL of 200 µM DPPH (Roth) with methanol (final concentration of 100 µM). The absorbance was read at 517 nm against a blank. The negative control was a mixture of the DPPH reagent and the solvent used. For each sample, the measurement was performed three times. The values were expressed as the mean ± SD. The percentage of DPPH free radical reduction was calculated using the following equation: % DPPH reduction = (A − Ax)/A × 100, where A is the absorbance of the DPPH solution with the solvent and Ax is the absorbance of the DPPH solution with a sample solution. The DPPH scavenging activity was expressed as the concentration of the fraction required to decrease the DPPH absorbance by 50% (IC_50_). The IC_50_ value was determined from the graph of the % DPPH reduction plotted against the sample’s final concentrations [44]. The antioxidant activity index (AAI) [45] was calculated from the equation: AAI = DPPH final concentration (µg mL^−1^)/IC_50_ (µg mL^−1^).

### 4.10. Statistical Analysis

Data were expressed as the means ± SE for the three independent experiments. The collected data were subjected to a one-way analysis of variance (ANOVA) followed by Duncan’s post hoc test. A two-sided *p*-value of 0.05 was used to declare statistical significance. All analyses were conducted using STATISTICA v. 10 (StatSoft, Inc., Tulsa, OK, USA, 2011).

Data from the PCR-array analysis were expressed as the means ± SE for three independent experiments. Statistical analysis was performed with a one-way analysis of variance (ANOVA) followed by the Student–Newman–Keuls test. *P*-values less than 0.05 were considered statistically significant.

## 5. Conclusions

To the best of our knowledge, the current study is the first to demonstrate that a saponin fraction from the roots of *Eryngium planum* L. was able to regulate the expression of genes mostly specific to the mitochondria-related apoptosis pathway. Some authors have reported that several naturally occurring anti-tumor saponins cause apoptosis via the mitochondrial pathway.

## Figures and Tables

**Figure 1 plants-12-02485-f001:**
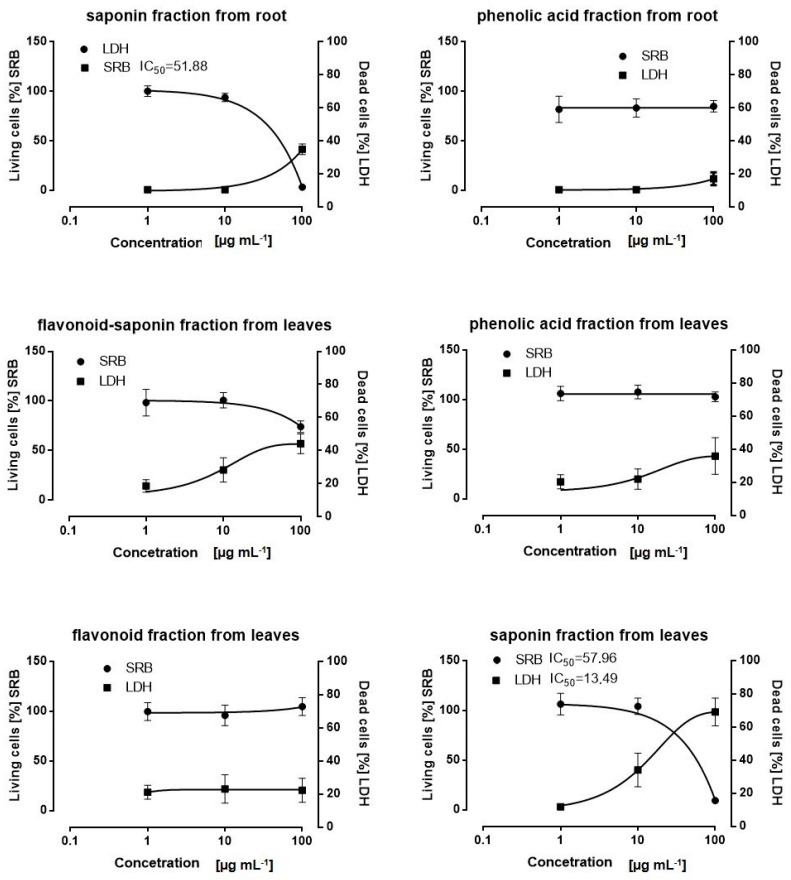
Effect of fractions from the roots and leaves of *Eryngium planum* L. on SKOV-3 cell viability. After 24 h treatment with fractions at the range of concentration 1–100 µg mL^−1^, IC_50_ values were determined by SRB and LDH assays. The results of three independent replicates are presented as the mean ± SEM.

**Figure 2 plants-12-02485-f002:**
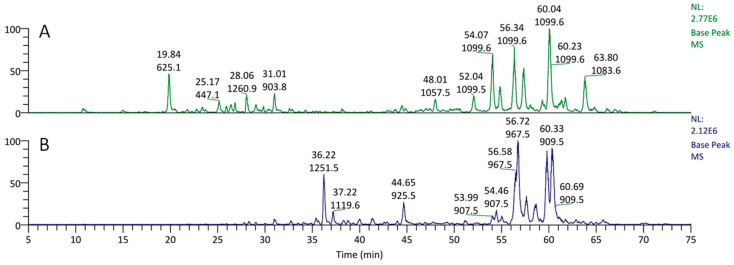
Chromatograms of methanolic fractions from (**A**) the rosette leaves and (**B**) the roots of *Eryngium planum* L. (values: upper—retention time; lower—molecular ion mass).

**Figure 3 plants-12-02485-f003:**
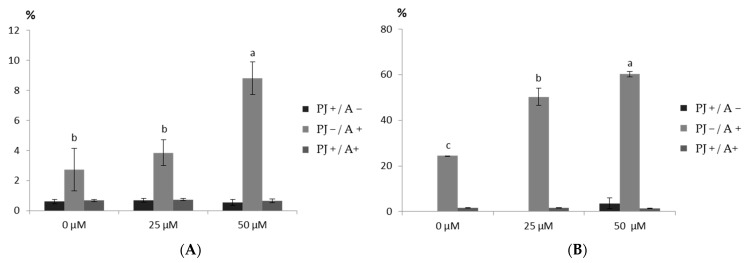
Effect of the saponin fraction from (**A**) the rosette leaves and (**B**) the roots of *Eryngium planum* L. on apoptosis induction in the SKOV-3 cells. The mean values with the same letter are not significantly different at *p* = 0.05 using Duncan’s multiple range test.

**Figure 4 plants-12-02485-f004:**
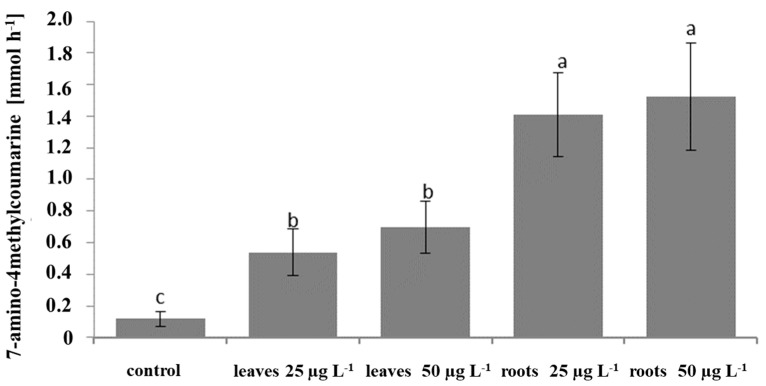
Effect of the saponin fractions from the leaves and roots of *Eryngium planum* L. on the caspase 3 activity in the SKOV-3 cells. Mean values with the same letter are not significantly different at *p* = 0.05 using Duncan’s multiple range test.

**Figure 5 plants-12-02485-f005:**
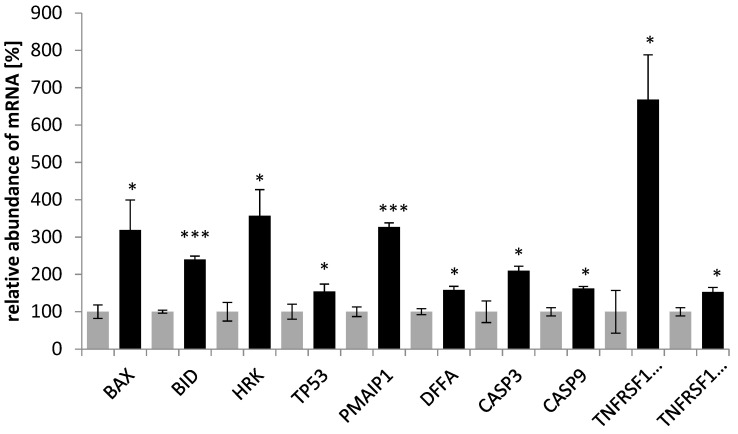
Effect of the saponin fraction from the roots of *Eryngium planum* L. on proapoptotic genes in the SKOV-3 cells. The PCR-array was used to analyze the level of indicated genes, treated for 24 h with the saponin fraction from the roots. The results of the three independent replicates are presented as the mean ± SEM *** *p* < 0.001 and * *p* < 0.05 compared to the control (gray—control; black—after treatment).

**Figure 6 plants-12-02485-f006:**
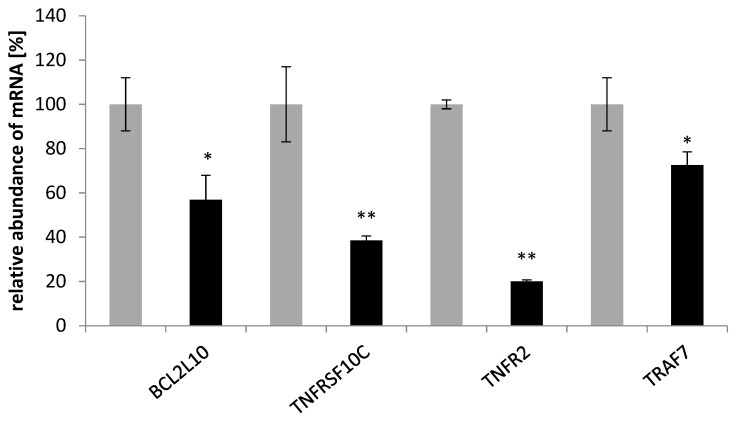
Effect of the saponin fraction from the roots of *Eryngium planum* L. on antiapoptotic genes in the SKOV-3 cells. The PCR-array was used to analyze the level of indicated genes, treated for 24 h with the saponin fraction from the roots. The results of the three independent replicates are presented as the mean ± SEM ** *p* < 0.01 and * *p* < 0.05 compared to the control (gray—control; black—after treatment).

**Table 1 plants-12-02485-t001:** The names and chemical structures of the three main triterpenoid saponins occurring in *Eryngium planum* L.

Triterpenoid Saponin Name	Compound Formula	Ref.
3-*O*-β-D-glucopyranosyl-(1→2)-β-D-glucuronopyranosyl-21-*O*-acetyl-22-*O*-angeloyl-R1-barrigenol	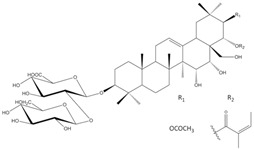	[6,8]
3-*O*-β-D-glucopyranosyl-(1→2)-β-D-glucurono-pyranosyl-22-*O*-angeloyl-A1-barrigenol	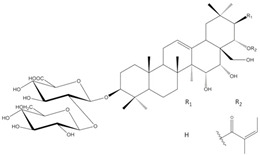	[6,8]
3-*O*-β-D-glucopyranosyl-(1→2)-β-D-glucuronopyranosyl-22-*O*-angeloyl-R1-barrigenol	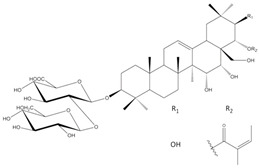	[6,8]

**Table 2 plants-12-02485-t002:** Content (mg g^−1^ DW) of triterpenoid saponins (ES1–ES6) in the methanolic extracts from the leaves and roots of *Eryngium planum* L.

Plant Material	Saponins
	ES2 + ES3
Rosette leaves	0.185 ± 0.01
	ES1 + ES2 + ES3 + ES4 + ES5 + ES6
Roots	5.626 ± 0.02

**Table 3 plants-12-02485-t003:** The DPPH (100 µM) scavenging activity (mean ± SD) of the column chromatography fractions of the extracts from the leaves and roots of *Eryngium planum*.

Column Chromatography Fraction (Fraction Number)	% DPPH Reduction (100 µg mL^−1 a^)	IC_50_ Value ^b^ (µg mL^−1^)	AAI Value ^c^
Leaves			
Flavonoid fr. (9–19)	28.99 ± 0.21	-	-
Flavonoid-saponin fr. (20–22)	23.64 ± 0.18	-	-
Saponin fr. (23–29)	36.82 ± 0.07		
Phenolic acids fr. (30–33)	62.37 ± 0.13	76.81 ± 0.18	0.51 ± 0.01
Roots			
Saponin fr. (6–14)	0	-	-
Phenolic acids fr. (15–16)	80.84 ± 0.08	34.55 ± 0.05	1.14 ± 0.02
Reference			
Quercetin	95.78 ± 0.05	2.70 ± 0.01	14.58 ± 0.03

^a^ at the highest concentration measured; ^b^ IC_50_—Inhibitory concentration 50%; ^c^ AAI—antioxidant activity index.

## Data Availability

Not applicable.

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
