# Peer review of "Saponin Fractions from Eryngium planum L. Induce Apoptosis in Ovarian SKOV-3 Cancer Cells"

_plants, 2023, doi:10.3390/plants12132485_

Round 1
Reviewer 1 Report
The authors did a thorough investigation on the evaluation of different fractions from Eryngium planum L. This article compared the apoptosis induction by different extract fractions from plant roots or leaves and indicated that the saponin fractions from the roots showed the best cytotoxicity on the SKOV-3 cell line. In addition, the authors tested the expression changes of pro-apoptotic and anti-apoptotic genes in the SKOV-3 cell line upon treatment to reveal the potential pathway behind the induced apoptosis. Overall, the manuscript did a comprehensive exploration regarding the characterization and evaluation of the saponin fractions from Eryngium planum L and will attract wide interest in this field. Here are some comments to the authors.
1. Could the authors provide the identification and structure at least for the major components in the saponin fractions from the roots?
2. In the method section, the authors mentioned that Annexin V was measured after 4 hrs of treatment. Is this incubation time long enough to reflect the full apoptosis induction?
3. Could the authors have the cytotoxicity data in normal ovarian cell lines as a control?
4. As the manuscript focused on apoptosis induction, I suggest the authors to provide more insight regarding the apoptosis pathway in this study which could be achieved by additional experiments measuring mitochondrial membrane potential, calcium influx, or cytochrome c release.
5. Is there a relevant animal study performed? Could the authors provide some preliminary data or discussion from literature to illustrate the fate of the extract fractions in the circulation system? Is there any off-target toxicity observed and how to avoid it?
Reviewer 2 Report
The manuscript (Saponin fractions from Eryngium planum L. induce apoptosis 2 in ovarian SKOV-3 cancer cells) - plants-2438120 has been reviewed.
The aim of the study is clair
The results well presented and discussed
However, the introduction section should be improved by adding new data as the citations are relatively old. and should be updated.
The structure activity of saponins-cytotoxic activity should be discussed.
Round 2
Reviewer 1 Report
Accept in present form